# Lysophosphatidylcholine: Potential Target for the Treatment of Chronic Pain

**DOI:** 10.3390/ijms23158274

**Published:** 2022-07-27

**Authors:** Jinxuan Ren, Jiaqi Lin, Lina Yu, Min Yan

**Affiliations:** Department of Anesthesiology, Second Affiliated Hospital, School of Medicine, Zhejiang University, Hangzhou 310009, China; 12018551@zju.edu.cn (J.R.); 21918625@zju.edu.cn (J.L.); zryulina@zju.edu.cn (L.Y.)

**Keywords:** chronic pain, lysophosphatidylcholine, metabolism, lipidomics, biomarkers, G protein-coupled receptors, Toll-like receptors, ion channels

## Abstract

The bioactive lipid lysophosphatidylcholine (LPC), a major phospholipid component of oxidized low-density lipoprotein (Ox-LDL), originates from the cleavage of phosphatidylcholine by phospholipase A2 (PLA2) and is catabolized to other substances by different enzymatic pathways. LPC exerts pleiotropic effects mediated by its receptors, G protein-coupled signaling receptors, Toll-like receptors, and ion channels to activate several second messengers. Lysophosphatidylcholine (LPC) is increasingly considered a key marker/factor positively in pathological states, especially inflammation and atherosclerosis development. Current studies have indicated that the injury of nervous tissues promotes oxidative stress and lipid peroxidation, as well as excessive accumulation of LPC, enhancing the membrane hyperexcitability to induce chronic pain, which may be recognized as one of the hallmarks of chronic pain. However, findings from lipidomic studies of LPC have been lacking in the context of chronic pain. In this review, we focus in some detail on LPC sources, biochemical pathways, and the signal-transduction system. Moreover, we outline the detection methods of LPC for accurate analysis of each individual LPC species and reveal the pathophysiological implication of LPC in chronic pain, which makes it an interesting target for biomarkers and the development of medicine regarding chronic pain.

## 1. Introduction

Chronic pain is a common, complex, and distressing problem [1], which is characterized by persistent pain even after the initial irritating injury/event has subsided [2], and it has significant societal and personal implications [1]. It affects more than 20% of adults in developed nations. In the U.S. alone, the direct and indirect costs exceed $600 billion annually. Additionally, the experience of chronic pain begins early; as many as 38% of children and adolescents in the community sample have reported chronic pain [3]. It is usually caused by injury or disease; however, it is a separate condition in its own right, not just a symptom accompanying other diseases [1]. Poor management of severe chronic pain, possibly due to an imbalance between analgesics and tolerability, is a burden for patients, with side effects that often lead to discontinuation of treatment [4]. In recent years, interventions for chronic pain are still not completely satisfactory, probably due to the variety of persistent pain conditions with different pathological processes, such as musculoskeletal [5], neuropathic [6], visceral [7], and cancer-related [8] pain, whose pathophysiological mechanisms have not been completely explored.

One mechanism underlying the development and maintenance of chronic pain is oxidative stress [9]. Reactive oxygen species (ROS) have been identified as key factors in nearly all human diseases, including chronic and acute diseases such as atherosclerosis, chronic pain, and acute lung/liver/kidney injuries [10]. The initiation of lipid peroxidation begins with the interaction between polyunsaturated fatty acids and reactive oxygen species [10]. Increased ROS and especially lipid peroxidation are implicated in the pathogenesis of several chronic pain diseases. In the fibromyalgia model, animals were accompanied by increased oxidative stress and lipid peroxidation [11]. Oxidative stress damage was shown to be one of the important factors that induced neuropathic pain [12]. Rats with osteoarthritis had a high level of malonaldehyde (MDA) (lipid peroxidation marker) production [13]. Reactive oxygen species and lipid peroxidation inhibitors reduced mechanical sensitivity in chronic pain models [14]. Compounds such as Apocynin, an NADPH oxidase inhibitor, limited the production of ROS precursor superoxide to reduce ROS, which inhibited inflammation in animal models of nerve tissue damage. The efficacy of 4-oxo-tempo may be related to the effects of the direct scavenging of oxidative radicals in animals with a chronic neuropathic pain model [15]. Antioxidants (N-acetylcysteine and Tempol) significantly reduced oxidative stress in the serum (assessed by MDA and H_2_O_2_ levels) of mice with stress-related chronic pain disorders [16]. Importantly, lysophosphatidylcholine (LPC) is an endogenous product derived from peroxidation during oxidative stress [17]. In response to lipid peroxidation from inflammation and tissue injury, phospholipids undergo lipid peroxidation to LPC [18]. Exposure to endogenous and exogenous LPC has emerged as a key contributor to cellular and tissue biology, such as inflammatory cascades [19] in chronic disease states—for example, diabetes, cancer, cardiovascular diseases, or neurodegeneration [20,21,22,23]. The development and maintenance of human chronic pain diseases have possibly established a causal link with specific LPC [24]. Although the clinical and pathological manifestations of chronic pain are broad, inflammation covers all stages of the disease, and various bioactive lipids have been implicated in such inflammation in various cells, highlighting their involvement in the pain transduction process [25]. Following inflammation, the excitatory neurotransmitter substance P and glutamate are released from primary afferent neurons, promoting the synthesis of lysophosphatidylcholine (LPC) [26]. In addition, injury to nervous tissue leads to an increase in reactive oxygen species (ROS) and promotes the synthesis of LPC, which enhances the membrane hyperexcitability to induce chronic pain. Antioxidants also effectively prevent the synthesis of lipid LPC and alleviate the symptoms of chronic hyperalgesia in animal models [16]. The purpose of this article is to summarize what is known about LPC, including its function and related signal regulation pathways in chronic pain diseases.

## 2. Lysophosphatidylcholine (LPC)

### 2.1. The Metabolism and Species of LPC

LPC, an important lipid molecule in mammalian tissues, belongs to a group of bioactive lysophospholipids [27]. Molecular species of LPC are identified by the lengths and saturation of their acyl chains. LPCs are produced from cell-membrane-derived phosphatidylcholine (PC) as a result of hydrolysis by phospholipases [28,29] (Figure 1). Two phospholipases have been studied, namely secretory PLA2 (sPLA2) and lipoprotein-associated PLA2 (Lp-PLA2) [30]. sPLA2 is Ca^2+^-dependent and hydrolyzes the sn-2 acyl group of the glycerophospholipids in lipoproteins and cell membranes to yield LPC and free fatty acids. In contrast to sPLA2, Lp-PLA2, also known as platelet-activating factor (PAF)-acetylhydrolase (PAF-AH) is Ca^2+^-independent, and it is specifically for short acyl groups at the sn-2 position of the phospholipid substrate. Lp-PLA2 can also hydrolyze oxidized phospholipids to generate LPC and oxidized fatty acids. LPC is usually present in very small concentrations because of LPC catabolism through different pathways mediated by separate enzymes: (1) after synthesized, LPC is secreted outside the cell and hydrolyzed to lysophosphatidic acid (LPA) and choline by autotaxin (ATX) [31]; (2) LPC is converted back to PCs by the enzyme lysophosphatidylcholine acyltransferase (LPCAT) in the presence of Acyl-CoA [32]; (3) LPC molecules catalyzed by cytosolic lysophospholipase-transacylase (LPTA) to form PC and glycerophosphorylcholine (GPC) [33] (Figure 1). The accumulation of LPC reflects increased PLA2-catalyzed PC hydrolysis or decreased LPC catabolism or a combination of both processes [34]. 

In recent years, research has begun to focus on the accurate analysis of each individual LPC species. Various LPC species have been identified by specific detection methods according to carbon chain length and number of double bonds [35], including LPC(14:0), LPC(15:0), LPC(16:0), LPC(16:1), LPC(17:0), LPC(18:0), LPC(18:1), LPC(18:2), LPC(18:3), LPC(20:0), LPC(20:2), LPC(20:3), LPC(20:4), LPC (22:6), LPC(26:0), LPC(28:1), and so on [36,37,38,39,40]. As a pro-inflammatory lipid, abnormal levels of LPC in body fluids such as blood, urine, synovial fluid, cerebrospinal fluid, and tissues are closely related to pathological states.

### 2.2. Detection Methods of LPC

The detection of LPC relies on the rise of lipidomics [16,41]. Lipidomics is a branch of metabolomics, and it is generally believed that lipidomics is a discipline that focuses on the qualitative and quantitative screening of metabolites in an organism and their roles in protein expression and gene regulation [42]. Lipidomic analyses have emerged based on existing omics disciplines and have developed rapidly in recent years [43] (Figure 2).

Lipidomics can assay metabolite compositions through various targeted and non-targeted techniques [44,45]. Prevailing technological advances have made accurate profiling of LPC in biological samples, such as nuclear magnetic resonance (NMR) spectroscopy [46,47,48,49], liquid chromatography coupled with mass spectrometry (LC-MS) [50,51,52], gas chromatography coupled to mass spectrometry (GC-MS) [51,53], high- or ultra-high-performance liquid chromatography coupled to UV or fluorescent detection (HPLC/UPLC) [54,55], and matrix-assisted laser desorption/ionization mass spectrometry (MALDI-MS) [56]. Each analytical platform has its own advantages and disadvantages (Table 1).

#### 2.2.1. NMR Spectroscopy

Nuclear magnetic resonance (NMR) spectroscopy, a technique for detecting the chemical environment of an atomic nucleus by absorbing radio frequency electromagnetic radiation [57], is an unbiased, nondestructive, and easily quantifiable form of sample processing, requiring little or no chromatographic separation and allowing for the routine identification of novel compounds. In addition, NMR is highly automatable, has extremely high reproducibility, and is feasible for high throughput [58]. NMR covers a wide range of applications, not limited to the analysis of biological fluids or tissue extracts. The nuclei best suited for NMR spectroscopy in biological systems include ^1^H, ^19^F, ^31^P, ^13^C, and ^15^N [59]. (^1^H) NMR spectroscopy (^1^H NMR) is commonly used in the profiling of LPC [60]. However, NMR spectroscopy has lower sensitivity and is suitable for the quantification of metabolites present in relatively high concentrations [57].

#### 2.2.2. LC-MS

Liquid chromatography coupled with mass spectrometry (LC-MS) is a combination of liquid chromatography and mass spectrometry [61]. LC-MS combines the separation capabilities of LC with the mass analysis power of MS [62]. The LC-MS detection method has the advantages of excellent resolution and sensitivity, small sample volumes, and relatively low costs, making it the most powerful analytical tool for metabolites today [63,64,65,66,67].

#### 2.2.3. GC-MS

Gas chromatography coupled with mass spectrometry (GC-MS) is also a commonly used platform for metabolomic research [68]. GC-MS was the first instrument used for metabolite profiling of human blood and urine by Horning in 1971 [69]. Apart from the high sensitivity and throughput [70,71], due to its longer use in clinical chemistry practice, GC-MS also possesses a higher chromatographic resolution and larger databases of identified peaks compared to the LC-MS. To some extent, GC–MS avoids the common problems of LC–MS, such as matrix effects and ion suppression by co-eluting compounds [72,73]. 

#### 2.2.4. HPLC/UPLC

The history of high-performance liquid chromatography (HPLC) can be traced back to the early 20th century [74]. Over the years, HPLC has made great progress in terms of speed, convenience, high sensitivity, choice of column stationary phase, suitability for various sample matrices, and the combination of chromatographic methods with spectral detectors [75,76]. It suffers from limitations such as low throughput, lack of high efficiency, inability to observe non-electrochemically active species, and difficulties associated with metabolite identification [77,78]. Ultra-performance liquid chromatography (UPLC) makes full use of chromatographic principles for separation, using short columns packed with smaller particles (sub-2 lm). Reduced analysis time, increased peak efficiency (peak width), better resolution, and reduced solvent usage are observed compared to conventional HPLC [79]. 

**Table 1 ijms-23-08274-t001:** Advantages and disadvantages of metabolomics techniques.

Method	Advantages	Disadvantages	References
NMR spectroscopy	Great range of detectable molecular species; Simple sample preparation; Excellent reproducibility; High automation	Low sensitivity; Quantification of relatively high concentrations of metabolites/extensive	[57,58]
LC-MS	High sensitivity; Small sample volumes; Relatively low costs; Superior resolution	Matrix effects and ion suppression by co-eluting compounds; Limitation of detectable metabolites	[63,64,65,66,67]
GC-MS	High chromatographic resolution; Large databases of identified peaks; High sensitive; High throughput	A large number of unidentified peaks; Require additional analytical steps; Separate and identify low molecular weight	[70,71,72,73]
HPLC	Robustness; Convenience; Good selectivity; High sensitivity	Low throughput; Inability to observe non-electrochemically active species; Difficulties of metabolite identification; Lack of high efficiency	[75,76,77,78]
UPLC	Short analysis time; Improved peak efficiency; Better resolution; Decreased use of solvents	Less time life of columns	[79]
MALDI-MS	Suitability for solid samples; High sensitivity; Easy sample handling; Salt tolerance; High speed	Limitation of detectable metabolites	[43]

#### 2.2.5. MALDI Mass Spectrometry

Matrix-assisted laser desorption/ionization mass spectrometry (MALDI-MS) is a powerful method for the simultaneous detection and identification of many molecules directly from biological samples of animals or humans. MALDI-MS can detect a variety of biomolecules, from small to large. Due to the broad applicability of this method, MALDI-MS is widely used in lipidomics or metabolomics studies [80]. The advantages of MALDI mass spectrometry include high sensitivity, easy sample handling, salt tolerance, rapid speed, and suitability for solid samples. However, it is selective for the detected lipid metabolites [43].

## 3. Lysophosphatidylcholine and Chronic Pain Diseases 

From numerous reports, it has been clarified that the level and metabolism process of LPC in the body fluid or tissues of animals or humans are elevated in various chronic pain states, such as chronic inflammatory pain [81], chronic joint pain [82,83] neuropathic pain [29,39,84], fibromyalgia [16], and multisite musculoskeletal pain [38] (Table 2). In this section, we summarize these LPC-induced responses and cellular mechanisms in detail.

### 3.1. Inflammatory Pain

Inflammatory pain is the most important clinical symptom of inflammatory diseases. The skin and joints are systems that are particularly susceptible to the formation of inflammatory pain. The common pathogenesis is that pro-inflammatory mediators such as chemokines, cytokines, growth factors, neuropeptides, and proteases are released at sites of inflammation and subsequently sensitize peripheral pain-sensing neurons [85]. In addition to the above molecules, lipids can also act as inflammatory modulators to induce inflammatory pain. Katelyn E Sadler et al. used LC-MS techniques to confirm that LPC was markedly elevated in mice with CFA-induced inflammatory pain in the skin; the content of LPC in the paw tissue of CFA-injected mice reached 130 μM, twice that of the vehicle-treated group. However, circulating LPC concentrations were unchanged in CFA-injected animals, suggesting that the excess lipid was derived from cells localized to the injured tissue. Furthermore, wild-type mice developed mechanical allodynia after dural injection of LPC [81], which reflected the correlation between LPC and CFA-induced inflammatory pain.

### 3.2. Chronic Joint Pain

Chronic joint pain is the main reason for patients to seek medical treatment for chronic pain; it seriously affects the quality of life of patients, resulting in disability and psychological distress [82]. Rheumatoid arthritis (RA) and osteoarthritis (OA) are associated with a risk of developing persistent chronic joint pain [83]. Florian Jacquot et al. demonstrated that LPC correlated with pain outcomes in a cohort of chronic joint pain patients. The synovial fluid levels of LPC in the 50 patients (32 women and 18 men) were evidently elevated, especially the LPC (16:0) species, compared with control subjects via high-definition mass spectrometer (HDMS). Intra-articular injection of LPC (16:0) resulted in persistent pain and anxiety-like behavior in mice, suggesting that LPC (16:0) could be considered a trigger for chronic joint pain in male and female mice [82]. Moreover, it has been demonstrated that mice injected with B02/B09 monoclonal antibodies (mAbs) isolated from B cells of patients with RA developed a long-term mechanical hypersensitivity accompanied by bone erosion and elevated LPC (16:0). In addition, elevated levels of LPC and sPLA2, a family of enzymes required for LPC synthesis, have been verified in the plasma and synovial fluid of patients with RA and OA, as well as those with joint pain. Consequently, it was possible that LPC was regarded as a biological target for predicting chronic joint pain in rodents or humans, especially LPC (16:0) (Figure 3) [83].

### 3.3. Fibromyalgia and Multisite Musculoskeletal Pain (MSMP) 

Fibromyalgia (FM) is characterized by chronic widespread musculoskeletal pain and associated fatigue, memory problems, and sleep disturbances [86,87]. Most of the lipidomics studies identified by our search were on this type of chronic pain. Chih-Hsien Hung et al. utilized untargeted lipidomic analysis and QqQ MS, respectively, to identify the serum and plasma of C57BL/6J mice and 31 fibromyalgia patients and 30 healthy controls at different time points. The identified lipids were mainly LPCs. LPCs (16:0) in the fibromyalgia mouse model were upregulated by 1.37-fold of the basal status in mice. It has also been proposed that central sensitization occurs after repeated intramuscular injections of LPC (16:0) in mice, which resulted in the activation of c-fos and pERK in spinal dorsal horn neurons [16]. Increased LPC (16:0) expression in FM patients also correlated with pain symptoms [16]. In addition to LPC (16:0), LPC (18:1) was also increased in the fibromyalgia mouse model. This may partly explain the increasing prevalence of fibromyalgia in the female population [88]. Wei-Hsiang Hsu et al. revealed several potential biomarkers of FM mice, some not previously described, such as LPC (20:3) in serum via ^1^HNMR-and LC-MS-based metabolomics profiling [50]. In addition, LPC (16:0) in the serum was also upregulated, which was the same result as in the study of Chih-Hsien Hung. Pierluigi Caboni et al. showed, using a metabolomics approach combining liquid chromatography-quadrupole-time of flight/mass spectrometry (LC-Q-TOF/MS) with multivariate statistical analysis, that lipid compound LPCs were elevated in the plasma of 22 females affected by FM and 21 controls [89]. In addition, in a large targeted metabolic profiling study, the metabolites were measured in the plasma of 122 non-multisite musculoskeletal pain (MSMP) and 83 MSMP patients. This study demonstrated that two lysophosphatidylcholines, LPC (26:0) and LPC (28:1), were significantly upregulated and positively associated with MSMP [38]. 

**Table 2 ijms-23-08274-t002:** The Application of LPC in Chronic pain.

Year	Author	Disease	Samples	Method	Observations	References
2021	Katelyn E Sadler et al.	CFA-induced inflammatory pain; skin incision-induced pain; chemotherapy-induced peripheral neuropathic pain	Mice hindpaw skin	LC-MS	CFA induced inflammatory pain, skin incision, and chemotherapy-induced peripheral neuropathy, all of which were characterized by elevated concentrations of LPC.	[81]
2022	Florian Jacquot et al.	Chronic joint pain	Synovial fluids from 50 patients (32 women and 18 men)	HDMS	The synovial fluid levels of LPC were significantly elevated, especially the LPC (16:0) species, compared with postmortem control subjects.	[82]
2021	Alexandra Jurczak et al.	B02/B09-induced pain	Bone marrow extracts of B02/B09-treated mice	HDMS	LPC (16:0) was the most abundant and significantly increased in the B02/B09 group compared with control.	[83]
2020	Chih-Hsien Hung et al.	Fibromyalgia	Serum from RISS mice; plasma from 31 fibromyalgia patients and 30 healthy controls	Untargeted lipidomic analysis/QqQ MS	LPC (16:0) in fibromyalgia mouse and patients were upregulated.	[16]
2019	Wei-Hsiang Hsu et al.	Fibromyalgia	Mice serum	^1^H NMR and LC-MS	Impactful metabolites in the FM model including LPC (16:0), LPC (20:3) in serum.	[50]
2014	Pierluigi Caboni et al.	Fibromyalgia	Plasma from 22 females FM patients and 21 controls	LC-MS	Plasma of FM patients identified many lipid compounds, mainly including LPC.	[89]
2021	Ming Liu et al.	Multisite musculoskeletal pain (MSMP)	Plasma of 122 non-MSMP and 83 MSMP patients	Biocrates AbsoluteIDQ p180 kit	LPC (26:0) and LPC (28:1) are associated with MSMP.	[38]
2021	Baasanjav Uranbileg et al.	Cauda equina compression	CSF and plasma from CEC rats; CSF from lumbar spinal canal stenosis patients and controls	LC-MS/MS; UHPLC-MS/MS	Lots of LPC species were significantly increased, especially LPC (16:0), LPC (18:2), LPC (20:4).	[39]
2020	Vittoria Rimola et al.	Oxaliplatin-induced Peripheral Pain	Mice sciatic nerve, DRG, dorsal spinal cord	LC-MS/MS	LPC (18:1) and LPC (16:0) were significantly increased after oxaliplatin treatment.	[29]
2011	Jun Nagai et al.	Partial sciatic nerve injury (SCNI)	Mice spinal cord and dorsal root	NALDI-MS	The levels of LPC (16:0), LPC (18:0) and LPC (18:1) were increased after SCNI.	[84]

HDMS: High-Definition mass spectrometer: LC-MS: Liquid chromatography mass spectrometry; ^1^H NMR: ^1^H-nuclear magnetic resonance; NALDI-MS: Matrix-assisted laser desorption/ionization mass spectrometry.

### 3.4. Neuropathic Pain

Neuropathic pain caused by a lesion or disease of the somatosensory nervous system is a common chronic pain condition and brings a lot of problems to humans [90]. The efficacy of current therapeutic drugs is limited, and it is essential to develop novel targets that permanently reduce or eliminate neuropathic pain [91]. In recent years, studies have demonstrated that metabolites are involved in the occurrence and development of neuropathic pain [92], and lipid LPCs are screened out. This is due to the fact that, following a nerve injury, the excitatory neurotransmitters substance P and glutamate are released from primary afferent neurons, or the increase in reactive oxygen species (ROS) leads to the upregulated synthesis of LPC [26]. Currently, pain induced by LPC injected into the median nerve has been regarded as a neuropathic pain model in many articles owing to pathological mechanisms of LPC-induced demyelination of the nervous system, which is inconsistent with the mechanism of LPC-mediated chronic joint pain [93,94,95]. Local LPC application results in the focal demyelination of afferent A fibers without axonal damage or loss of neurons in the dorsal root ganglia (DRG) [96]. In the central nervous system (CNS), LPCs also trigger a rapid demyelination without damage to adjacent cells and axons. This was thought to be a key role of immune cells in LPC-induced demyelination [97]. Peripheral macrophage and central microglia, as resident macrophages, contribute to maintaining homeostasis in the nervous system. Macrophages or microglia are activated in response to noxious stimuli such as nerve injury. Activated macrophages or microglia result in the production and release of pro-inflammatory mediators, which lead to the development of chronic pain [98,99]. LPC induced macrophage and microglia recruitment and activation in the mouse spinal cord [100,101,102]. In the LPC-induced model of demyelination, macrophages and microglia were detected at 48 h, when clear evidence of demyelination was observed [101,103]. Interestingly, the application of LPCs in the early presented a rapid but brief influx of T cells, and neutrophils, T cells, and neutrophils were seen in the spinal cord for 6–12 h [103]. This is because LPC caused rapid and extensive disruption of the blood–brain barrier, which induced early and transient T cell and neutrophil responses in the spinal cord. These cells likely promote a rapid influx of monocytes, followed by the activation of macrophages from monocytes and microglia to mediate demyelination [101,103]. In addition, LPC-induced demyelination induces mechanical allodynia and thermal hyperalgesia, which persists for at least 7 days (Table 3) [104,105]. LPC injection increased the levels of pain-related proteins, including neuropeptide Y (NPY), Nav 1.3, Nav 1.8, chemokines, and their receptors, in the DRG or spinal cord [93]. In the mice model of chemotherapy-induced peripheral pain, LPC (16:0) and LPC (18:1) were significantly increased in the sciatic nerve and DRG tissue, as revealed by untargeted and targeted lipidomics. Importantly, pain-like performance induced by LPC (16:0) and (18:1) was dependent on Ca^2+^ transients in primary sensory neurons [29]. Jun Nagai et al. developed a quantitative mass spectrometry assay to simultaneously analyze several species of LPCs in the SCNI. They found that the levels of LPC (16:0), LPC (18:0), and LPC (18:1) in the spinal cord and DRG were maximally increased [84]. Cauda equina compression (CEC) is a major cause of neurogenic claudication and progresses to neuropathic pain [39]. A study utilizing LC-MS/MS and UHPLC-MS/MS in rats and patients demonstrated that many LPC species were significantly elevated in the CSF and plasma of CEC model rat or CSF of patients with lumbar spinal canal stenosis (LSS), especially LPC (16:0), LPC (18:2), and LPC (20:4) (Figure 3). However, LPC levels in the spinal cord tissue samples of rats did not change dramatically [39].

### 3.5. The Enzymatic Pathways of Lysophosphatidylcholine (LPC) and Chronic Pain

In addition to the accumulation of LPC causing pain symptoms, molecules in the enzymatic pathways of LPC synthesis and catabolism, such as lysophosphatidic acid (LPA), autotaxin (ATX), and lysophosphatidylcholine acyltransferase (LPCAT), also play an important role in chronic pain. Accumulating evidence has revealed that LPC regulates the participation of platelet-activating factor (PAF)/PAF receptor (PAFr) in pain signal transduction [106]. LPC is hydrolyzed by autotaxin into LPA and acts through LPA receptors present on nociceptors. LPA, a potent bioactive lipid mediator, induces neuropathic pain as well as demyelination and pain-related protein expression changes via LPA receptor signaling [104]. Direct intrathecal administration of LPA was able to induce chronic pain responses in rodents [107,108]. LPA altered the density and activity of Ca11, K1, and TRP ion channels in microglia and neurons, causing allodynia and hyperalgesia, which played a central role in the initiation and maintenance of neuropathic pain [38]. Autotaxin mediated LPC to produce LPA, a bioactive lipid mediator that signals the activation of six GPCRs (LPA receptors 1-6). Autotaxin levels in synovial fluid and plasma correlated with disease severity in patients with knee OA [108]. Intrathecal LPC-induced mechanical allodynia and thermal hyperalgesia were significantly reduced in autotaxin heterozygous animals, indicating reduced conversion of LPC to LPA [104]. Moreover, ATX inhibition could ameliorate neuropathic pain symptoms by using ATX inhibitor (ONO-8430506) [109]. In addition, a recent study demonstrated that nerve injuries induced the production of LPA by converting LPC to LPA under the action of ATX, which was observed only in the spinal dorsal horn, but not in the spinal nerve, sciatic nerve, or DRG, for several hours. Furthermore, injury-induced synthesis of LPC and subsequent conversion to LPA were both involved in the development of neuropathic pain. However, injury-induced neuropathic pain and LPA production were attenuated to approximately 50% in atx^+/−^ mice and abolished in Lpar1^−/−^ mice, which was also observed in LPC-induced demyelination [110]. Therefore, the conversion of LPC to LPA may also be an important target for the treatment of chronic pain [111]. Of course, not all LPC will eventually be converted into LPA. The indications of the increased expression of the LPC to LPA-converting enzyme autotaxin or LPA receptors were not found in several chronic pain models [83]. This is because pathological pain is a complex state that may be related to both the model and the time of onset. Apart from the LPA, cyclic phosphatidic acid (cPA), produced from LPC using ATX, has a structure similar to that of LPA [112,113]. Unlike the biological function of LPA, cPA has the potential for use in the treatment of acute and chronic pain diseases because of its biological properties of anti-inflammatory and neuroprotective activities [112]. The cPA and its stable analog 2-carba-cPA (2ccPA) inhibited chronic and acute inflammation-induced C-fiber stimulation. The administration of 2ccPA significantly attenuated mechanical allodynia and thermal hyperalgesia following the partial ligation of the sciatic nerve, whether pretreatment or repeated post-treatments [114]. Intra-articular injection of 2ccPA also reduced the pain response to OA and articular swelling [115]. LPC is hydrolyzed by autotaxin into LPA and cPA, but the effects are completely opposite, suggesting that it may be related to the period and condition of synthesis. It has been suggested that LPC could be converted into cPA by HCl in a dose-dependent manner [116]. In addition, LPCAT is also promising as a novel therapeutic target for newly classified analgesic drugs. Hideo Shindou et al. confirmed that pain-like behaviors induced by partial sciatic nerve ligation (PSNL) were largely relieved by the deficiency of LPCAT [117].

## 4. LPC-Related Receptor and Chronic Pain

As mentioned above, the biological effects of LPC have been studied in mice and humans and are important in chronic pain. LPC acts as the ligand and can activate G protein-coupled receptors (GPCRs), Toll-like receptors (TLRs), and several ion channels, implicating the possible molecular mechanisms in the observed effects of LPC (Figure 4). 

### 4.1. LPC and G Protein Coupled Receptors

LPC is considered the ligand for GPCR, G2A (GPR132), and GPR4, with a significantly higher affinity for G2A than that for GPR4 [118,119]. LPC plays a key role in the development of chronic inflammatory diseases through the role of the G2A receptor. T cells overexpressing G2A exhibit chemotaxis to LPC; siRNA silencing in mouse T cell hybridomas and retroviral overexpression of G2A demonstrated the requirement for G2A in LPC-induced T cell migration [120]; G2A was also required for LPC-induced chemotaxis of macrophages [121], both demonstrating the interaction between LPC and the G2A effect. LPC led to an increase in intracellular calcium levels by acting on receptor G2A, resulting in increased neuronal excitability and activation of ERK mitogen-activated protein kinase (Figure 4). The signaling lipid receptor G2A and ERK mitogen-activated protein kinase were upregulated in a spared nerve injury (SNI)-induced neuropathic pain model [122,123]. There are few studies focusing on both LPC and GPR4, and the involvement of this mechanism in chronic pain has not been confirmed. Previous research has shown that LPC is associated with NLRP3 inflammasome and the release of IL-1β by GPR4 [124]. In the neurodegeneration and demyelination states, LPC activates NLRP3 inflammasomes in astrocytes and microglia [23], and NLRP3 inflammasome is involved in inflammatory pain [125] (Figure 4). In fact, the direct or indirect effect of LPC and G2A or GPR4 is controversial. In addition, the LPC derivative also targeted four other G protein-coupled receptors, namely GPR40, GPR55, GPR119, and GPR120 [126,127]. Studies have indicated that the stimulatory effect of isoprenoid derivatives of LPC on Ca^2+^ signaling in MIN6 cells was GPR40-, GPR55-, GPR119-, and GPR120-dependent [127]. GPR40/GPR55 has been implicated in inflammatory pain and neuropathic pain [128,129], but studies on LPC and these receptors in chronic pain are lacking. 

### 4.2. LPC and Toll-like Receptors

TLRs are an important family of receptors involved in complex intercellular signaling networks that develop in the context of chronic pain [130,131]. TLRs can induce an innate immune response by recognizing various pathogen-associated molecular patterns (PAMPs), and the receptors that recognize these molecular structures were named as pattern recognition receptors (PRRs) [132,133]. To date, there are ten known functional TLRs (TLR1-10) in humans and twelve TLRs (TLR1-9, TLR11-13) in mice. TLR1-2, TLR4-6, and TLR10 are located on the cell surface, and TLR3, TLR7-9, and TLR11-13 are observed in intracellular compartments [134]. In addition to innate immunity, TLRs are also expressed in the periphery and in CNS cells, and are coupled with the activation of various non-neuronal cells (microglia, schwann cells and astrocytes) and neurons, thus causing the release of pro-inflammatory cytokines and thereby leading to the generation and maintenance of chronic pain [135]. TLRs play a key role in OA [136], neuropathic pain [137], chronic pelvic pain [138], opioid-induced hyperalgesia [139], and cancer pain [140]. TLR2, TLR4, TLR5, TLR3, TLR7, TLR9, etc., have been reported to contribute to persistent pain [131]. Among them, TLR2 and TLR4 have been proven to be major Toll-like receptors that to mediate LPC function. LPC activated pain markers, such as NF-κB, p38 MAPK, and JUN and cytokine production (IL-6, TNF-α) by combining the TLR2 and TLR4 receptors (Figure 4) [118]. Previous studies have reported that LPC (18:0) and LPC (18:1) were more potent than LPC (16:0) and LPC (14:0) in promoting cytokine secretion from TLR-primed cells [141]. 

### 4.3. LPC and Ion Channels

Multiple ion channels are involved in sensing and transmitting nociceptive information in the neurons of the peripheral and central nervous system [142]. LPC can also exert its biological functions by binding to acid-sensing ion channels (ASICs) and transient receptor potential (TRP) ion channels (Figure 4). Acid-sensing ion channels (ASICs) are proton-activated cation channels that are expressed in a variety of neuronal and non-neuronal tissues, encoding several subunits (ASIC1, ASIC2, ASIC3, and ASIC4). ASIC3, an important pain transducer, can be activated by LPC and potentiated by many pro-inflammatory mediators [143,144]. A recent study showed that certain LPCs, especially LPC (16:0), were able to directly activate ASIC3 channels to mechanical stimuli, resulting in altered mechanoneuronal responses of primary afferent neurons [16]. In the fibromyalgia model, LPC-induced chronic hypersensitivity was obviously inhibited in APETx2 (a selective ASIC3 antagonist)-treated mice. Similarly, chronic hyperalgesic changes in WT animals were also robustly improved in Asic3^-/-^ mice after repeated LPC injections [16]. LPC (16:0) drove sufficient peripheral inputs to generate spinal sensitization process via ASIC3 channels in the mouse model of OA-induced inflammatory pain [82,83].

In addition, LPCs, such as LPC (18:1), activated the ligand-gated calcium channels’ transient receptor potential V1 and M8 (TRPV1 and TRPM8) in primary sensory neurons to induce mechanical hypersensitivity in mice, which stimulated chemotherapy-induced peripheral pain [29]. Lipid mass spectrometry indicated tissue-specific increases in LPC in pain models, accompanying mechanical allodynia, neuronal mechanical hypersensitivity, and spontaneous pain, which could be inhibited with transient receptor potential canonical 5 (TRPC5) inhibitors. TRPC5 is also a target of LPC-induced chronic pain [81]. TRPC5 inhibitors have demonstrated analgesic effects in all of the following conditions with elevated LPC: fibromyalgia [89], rheumatoid arthritis [145], osteoarthritis [146], lumbar spinal stenosis [89,147], diabetes [148], and migraine [149]. In addition, TRPV4 in DRG sensory neurons was essential for intrathecally LPC-induced chronic pain [105]. The above findings provide new molecular insights into the mechanism by which LPC may affect the activation of cellular signaling pathways in chronic pain. G protein-coupled receptors (GPCRs), ion channels, and Toll-like receptors are involved in nociceptive signaling and are considered important pharmacological targets for existing or potential drugs. Apart from GPCRs, TLRs, and several known ion channels, there are other receptors that may directly bind to LPC or some molecules that indirectly interact with LPC. Therefore, future research should continue to focus on physiological and therapeutic approaches to inhibit the LPC signaling cascade.

## 5. Conclusions

By and large, the studies on LPC and chronic pain have been scant as compared to studies on other pathological states. The current findings have highlighted the critical contribution of LPC in CFA induced-inflammatory pain, chronic joint pain, neuropathic pain, fibromyalgia, and multisite musculoskeletal pain, including total LPC and LPC species and rodents and humans (Table 2). We found that LPC (16:0), LPC (18:0), and LPC (18:1) were currently the three most detected LPC species in chronic pain, among which LPC (16:0) was involved in the chronic pain caused by osteoarthritis and fibromyalgia, while LPC (18:1) was more studied in nerve-injury-induced neuropathic pain. This suggests that specific LPC species may reflect some chronic pain diseases; after all, the mechanisms of chronic pain are complex and different. At present, the accurate detection of LPC relies on metabolomics or lipidomics technology, which provides assays for exploiting the role of LPC in chronic pain. Apart from chronic pain, LPC or LPC species in body fluids such as blood, urine, cerebrospinal fluid, and tissues are uniquely or collectively related to cancer [150,151,152], diabetes [153,154,155,156], coronary atherosclerosis [157], Alzheimer’s disease [158,159], rheumatoid arthritis [83], COVID-19 [160], liver and kidney damage [161,162], etc. Whether LPC is necessary for other chronic pain conditions, such as cancer pain, has not been confirmed. In addition, LPC-related metabolites, such as ATX, PLA2, cPA, and LPA, also serve as therapeutic targets of chronic pain. As an inflammatory lipid, LPC can activate downstream signaling pathways by binding to G protein-coupled receptors, Toll-like receptors, and several ion channels. It is also necessary to further explore new receptors for LPC in the future.

## Figures and Tables

**Figure 1 ijms-23-08274-f001:**
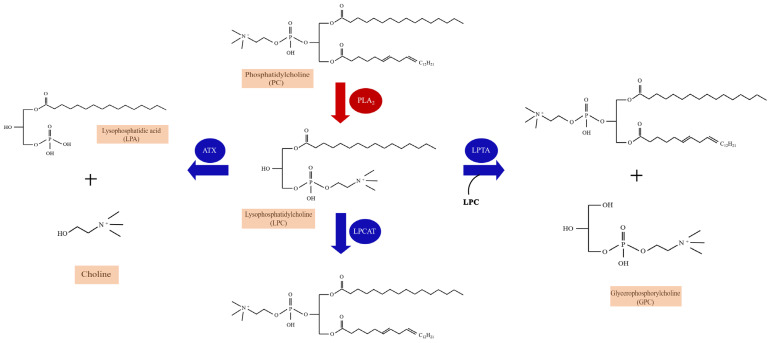
The enzymatic pathways of lysophosphatidylcholine (LPC) synthesis and catabolism. The production of LPC is the result of the fragmentation of the sn-2 residues of phosphatidylcholine (PC) hydrolyzed by PLA2. Three catabolism pathways of LPC are listed. LPC catabolism occurs through a disproportionation reaction involving two LPC molecules catalyzed by cytosolic lysophospholipase-transacylase (LPTA) to form PC and glycerophosphorylcholine (GPC). A hydrolytic pathway is catalyzed by autotaxin (ATX) to yield lysophosphatidic acid (LPA) and choline, and a reacylation pathway to form PC is catalyzed by lysophosphatidylcholine acyltransferase (LPCAT).

**Figure 2 ijms-23-08274-f002:**
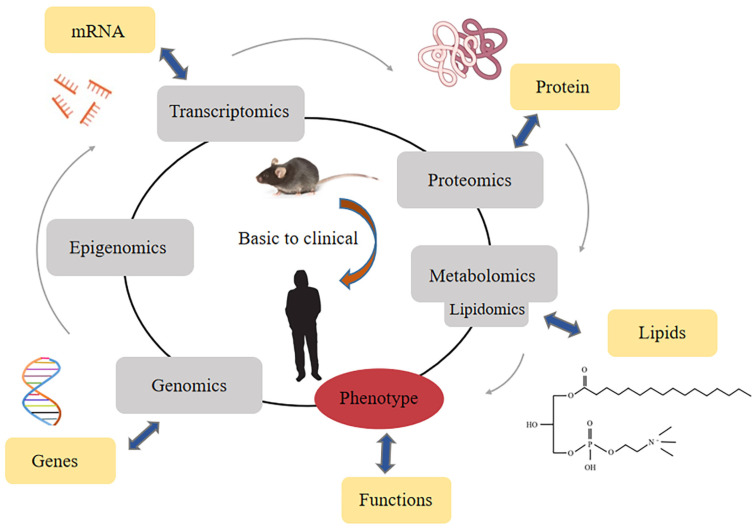
Classification of lipidomics in the area of all omics methods.

**Figure 3 ijms-23-08274-f003:**
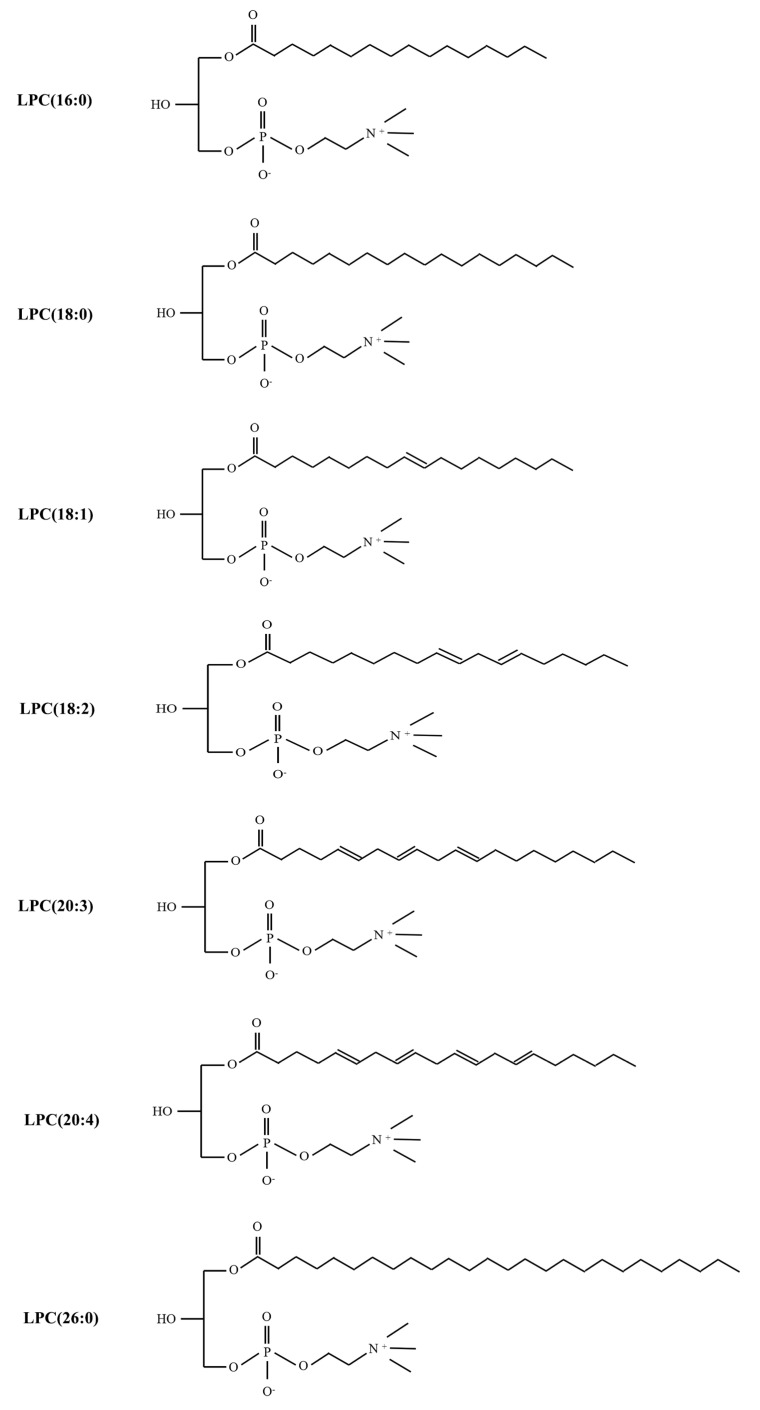
The structure of different LPC subtypes associated with chronic pain.

**Figure 4 ijms-23-08274-f004:**
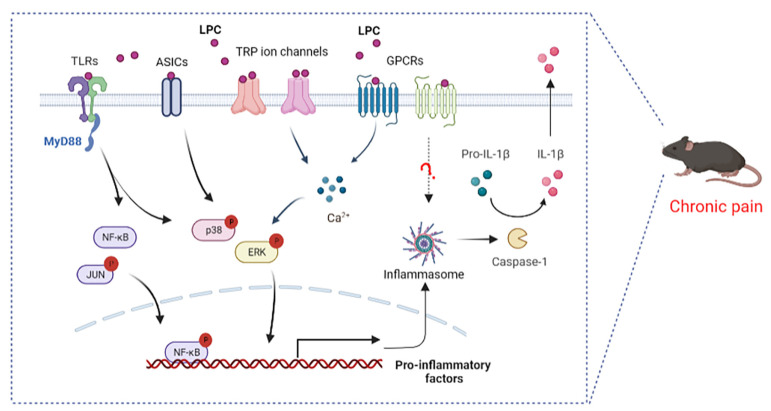
Proposed signaling pathways by which LPC mediates pain. LPC, lysophosphatidylcholine; TLRs, Toll-like receptors; GPCRs, G protein–coupled receptors; ASICs, acid-sensing ion channels; TRP, transient receptor potential; MyD88, myeloid differentiation factor; NF-κB, nuclear factor kappa-B; ERK, extracellular-signal-regulated kinase; Caspase-1, cysteinyl aspartate specific proteinase-1; IL-1β, interleukin-1 beta.

**Table 3 ijms-23-08274-t003:** The application of LPC in the construction of neuropathic pain models.

Year	Author	Administration	Species	Doses	Observations	References
2020	Chun-Ta Huang et al.	Intraneural injection	Sprague Dawley rats	4% LPC 2 μL	The rats developed mechanical allodynia and thermal hyperalgesia on day 1 after LPC treatment.	[94]
2021	Yong Chen et al.	Intrathecal injection	C57BL/6J mice	15 μg LPC	Intrathecal injection of LPC induced mechanical pain via activation of TRPV4-expressing DRG sensory neurons.	[105]
2013	Hsin-Ying Wang et al.	Intraneural injection	Male Wistar rats	4% LPC 2 μL	LPC treatment caused mechanic allodynia and thermal hyperalgesia.	[93]
2008	M Inoue et al.	Intrathecal injection	Male mutant mice	15 μg/50 μg LPC	A single injection of LPC at 15 μg showed significantly but slightly weaker mechanical allodynia on days 2–7. However, a higher dose of LPC (50 μg) caused abnormal behaviors.	[104]
2018	Hozo Matsuoka et al.	Intraneural injection	Wistar rats	2% LPC 5 μL	Paw withdrawal thresholds were significantly higher in the LPC group compared with the Non-LPC group.	[95]

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
