# Peer review of "Lysophosphatidylcholine: Potential Target for the Treatment of Chronic Pain"

_ijms, 2022, doi:10.3390/ijms23158274_

Round 1

Reviewer 1 Report

With their manuscript „ Lysophosphatidylcholine: potential target for the treatment of chronic pain”, the authors presented a summary of 139 articles about the effects of LPC on pain-related processes with a focus on detection methods, chronic pain diseases and  LPC receptors.  

LPC is a very important lipid metabolite which plays an important role in practically all physiological processes and diseases. Thus, it’s always very welcome to review LPC-effects and draw the respective conclusions and/or discuss its relation to various conditions in health and disease. However, to become a helpful article, selected aspect the authors mentioned in their manuscript must be discussed in a broader way to make sense.

·      Not all refvelant references are cited regarding LPC inducing pain being converted to LPA to activate the LPA receptors (e.g., DOI: 10.1186/1744-8069-6-78). Please review the literature more thorougfly

·      ATX is capable of producing not only LPA, but also cyclic phosphatidic acids (cPA) (PMID: 18560605). Importantly, cPA and its stable analog 2ccPA inhibit chronic and acute inflammation-induced C-fiber stimulation, and 2ccPA attenuates neuropathic pain (DOI: 10.1186/1744-8069-7-33) or reduces the pain response to OA (PMID: 25123228). So, add more information on cPA and pain

·      Spinal exposure to LPC can induce the expansion of T cells and the activation of macrophages/microglia, resulting in demyelination (DOI: 10.1002/glia.20449). Add more information on LPC and immune cells related to pain

·      The LPC derivative target not three but at least four GPCRs (additionally GPR120 (DOI: 10.3390/ijms22115748)

·      Correction of several typos is needed, e.g., 

Line 171 Lysophosphatidylcholine and Chronic pain diseases

Line 298 LPC-related receptor and Chronic pain

Reviewer 2 Report

In this review the authors detailed on LPC source, biochemical pathways and signal-transduction system. Moreover, they outline the detection methods of LPC for analysis of each individual LPC species and reveal the pathophysiological implication of LPC in chronic pain.

The review is well structured. 

I have minor consideration:

- The authors could add a role of antioxidants in the introduction

- The authors could add references in table 1

- Check for any spelling or space errors

- The role of TLR in pain should be better described

Round 2

Reviewer 1 Report

The authors changed the manuscript as required.